# Research Progress on Natural Small-Molecule Compounds for the Prevention and Treatment of Sepsis

**DOI:** 10.3390/ijms241612732

**Published:** 2023-08-12

**Authors:** Jingqian Su, Fen Zhou, Shun Wu, Zhiyong Tong

**Affiliations:** Fujian Key Laboratory of Innate Immune Biology, Biomedical Research Center of South China, College of Life Science, Fujian Normal University, Fuzhou 350117, China; 15859446856@163.com (F.Z.); qsx20221412@student.fjnu.edu.cn (S.W.); tongzhiyong1998@163.com (Z.T.)

**Keywords:** sepsis, small molecule compounds, inflammatory response, organ failure

## Abstract

Sepsis is a serious disease with high mortality and has been a hot research topic in medical research in recent years. With the continuous reporting of in-depth research on the pathological mechanisms of sepsis, various compounds have been developed to prevent and treat sepsis. Natural small-molecule compounds play vital roles in the prevention and treatment of sepsis; for example, compounds such as resveratrol, emodin, salidroside, ginsenoside, and others can modulate signaling through the NF-κB, STAT3, STAT1, PI3K, and other pathways to relieve the inflammatory response, immunosuppression, and organ failure caused by sepsis. Here, we discuss the functions and mechanisms of natural small-molecule compounds in preventing and treating sepsis. This review will lay the theoretical foundation for discovering new natural small-molecule compounds that can potentially prevent and treat sepsis.

## 1. Introduction

Sepsis is a serious disease with relatively high morbidity and mortality rates, involving life-threatening organ dysfunction caused by the host’s abnormal response to infection [1]. With the delisting of Eli Lilly’s (Indianapolis, IN, USA) Xigris in 2011, there is currently an absence of a specific pharmaceutical intervention for the management of sepsis; therefore, identifying effective drugs for the prevention and treatment of sepsis has become a research hotspot.

The pathogenesis of sepsis is complex. The initial acute response of the host to an invasive pathogen activates macrophages and produces a series of cytokines (pro-inflammatory and anti-inflammatory factors) that trigger apoptosis, necroptosis, and pyroptosis and activate damage-associated molecular patterns or pathogen-associated molecular patterns [2], which in turn trigger a cytokine storm [3]. A major change in the levels of cytokines in the body can cause damage to multiple organs, including the kidneys [4], lungs [5], liver [6], and heart [7], and eventually affect the immune function of the body and cause immune disorders. Inflammatory responses and immunosuppression occur sequentially during sepsis [8], and severe inflammatory responses in the body can lead to coagulation disorders [9]. Therefore, reducing inflammation, immunosuppression, and coagulation disorders are key parts of sepsis treatment. The drugs developed to regulate inflammatory responses include cytokine antagonists, pattern-recognition receptor inhibitors, recombinant human APC (Activated Protein C), and recombinant human soluble thrombosis regulators for regulating the blood coagulation system. Additionally, several anti-immunosuppressive drugs, such as cytokines (granulocyte colony-stimulating factor and granulocyte-macrophage colony-stimulating factor) and co-repressor molecule inhibitors, have been developed [10]. In addition to these drugs, this review will also focus on some natural medicinal substances, such as kombucha [11] and black mulberry [12], which exert antioxidant and anti-inflammatory effects in sepsis models and improve the immunity and survival rates of septic mice. Other small-molecule compounds, such as Fucoxanthin [13] and bis-N-norgliovictin [14], have also been reported to improve survival by reducing inflammation levels.

Natural small-molecule compounds (usually compounds with a molecular weight < 1000 Da) have attracted wide attention in drug research because of their characteristics and the advantages of rapid diffusion into cells to reach the target. Drugs can broadly be divided into several categories based on their structure and properties, such as polyphenols, anthraquinones, glycosides, flavonoids, and biogenic amines. Through a thorough investigation of the literature, this paper summarizes and discusses the functions and mechanisms of natural small-molecule compounds in preventing and treating sepsis.

## 2. Polyphenol Compounds

Polyphenols are compounds with multiple phenolic groups. Polyphenols can inhibit the activity of nuclear factor kappa-B (NF-κB), which in turn inhibits cell proliferation, angiogenesis, and metastasis and promotes apoptosis [15]. Currently, the main small-molecule polyphenols used for treating sepsis are resveratrol, curcumin, and tetrahydrocurcumin.

Resveratrol (CAS: 501-36-0) is a non-flavonoid polyphenol compound with anti-inflammatory, antiviral, antibacterial, and antitumor properties [16]. Thus far, research on the use of resveratrol for treating sepsis has mostly involved animal experiments; this compound has not yet entered the stage of clinical research regarding sepsis treatment. As shown in Figure 1, resveratrol can mitigate the acute kidney injury induced by sepsis. Sirtuin1 (SIRT1) is an NAD-dependent protein deacetylase, which is considered the main regulator of sepsis-induced acute kidney injury because it reduces oxidative stress and inflammation [17]. Activation of SIRT1 can inhibit the inflammatory response and oxidative stress. Resveratrol, as an SIRT1 activator, alleviates acute kidney injury in cecal ligation and puncture (CLP) septic mice by activating SIRT1 to promote deacetyl-mediated autophagy [18]. Resveratrol can also alleviate damage to other organs. For instance, it ameliorates the cardiomyocyte injury induced by lipopolysaccharide (LPS) by upregulating miR-149 and downregulating high mobility group protein B1 (HMGB1) [19]. Resveratrol also improves sepsis-associated encephalopathy by inhibiting the expression of the nucleotide-binding oligomerization domain, leucine-rich repeat, NOD-like receptor protein 3 (NLRP3), and interleukin-1β (IL-1β) [20] and can activate vascular endothelial growth factor-B (VEGF-B) expression and inhibit the NF-κB pathway to alleviate sepsis-induced acute lung injury [21]. However, high doses of resveratrol can increase intracellular oxidation, enhance mitochondrial membrane depolarization, and induce endothelial cell death [22].

Curcumin (CAS: 458-37-7) is a natural polyphenol compound with anti-inflammatory, anti-infection, and other biological properties [23]. As shown in Figure 2, curcumin alleviates lung injury in CLP model mice by regulating the differentiation of CD4^+^ T cells into Tregs, promoting the transformation of macrophages, and exerting anti-apoptosis, anti-inflammatory, and immunoregulatory effects [24]. Curcumin can also downregulate Toll-like receptor 1 (TLR1), inhibit the phosphorylation of NF-κB, and improve the survival rate of cardiomyocytes treated with lipopolysaccharides (LPS) [25]. Curcumin has also been reported to inhibit NF-κB and janus kinase2 (JAK2)/signal transducer and activator of Transcription 3 (STAT3) signaling and the expression of p-JAK2/STAT3, p-p65, and BCL2-Associated X (BAX) in mice with acute kidney injury to alleviate septic acute kidney injury effectively in CLP mouse models [26]. *Curcuma longa* extract is rich in turmeric. The main compound present in turmeric is β-turmerone (CAS: 19693-54-0). In a previous study, treatment with *Curcuma longa* extract had an anti-inflammatory effect and reduced the production of NO in an inflammation model induced by LPS [27]. In another study, targeted inhibition of toll-likereceptor4(TLR4) mediated downstream information transmission, thereby effectively preventing brain injury caused by neuroinflammation in LPS model mice [28].

After hydrogenation of curcumin, tetrahydrocurcumin (CAS: 36062-04-1) is obtained, which contains fewer unsaturated C double bonds in its structure; thus, tetrahydrocurcumin has higher stability, stronger antioxidation effects, and higher bioavailability than curcumin. As shown in Figure 3, tetrahydrocurcumin significantly increased the expression of SIRT1 and inhibited inflammation and oxidative stress, thereby preventing sepsis-induced acute kidney injury in a CLP mouse model [29].

## 3. Anthraquinone Compounds

Many natural anthraquinones have anticancer, anti-inflammatory, antioxidant, anti-osteoporosis, and other physiological properties [30]. The natural small-molecule anthraquinones used in sepsis treatment include emodin and aloin.

Emodin (CAS: 518-82-1) is a natural anthraquinone compound with numerous pharmacological effects, including anticancer, antiviral, anti-inflammatory, antibacterial, and hepatoprotective activities [31]. Thus far, research on the utility of emodin in preventing and treating sepsis has primarily focused on in vitro cell and animal experiments. Emodin can relieve lung injury, intestinal mucosal barrier injury, and cognitive impairment caused by sepsis through the scorch signaling pathway and the Vitamin D receptor (VDR)/NF-E2-related factor 2 (Nrf2) pathway. As shown in Figure 4, emodin alleviates NLRP3-induced acute lung injury by inhibiting LPS-dependent scorch death signaling [32]. It is known that VDR can activate the SIRT1/Nrf-2 pathway [33]. Emodin inhibits SIRT1-mediated HMGB1 protein expression by increasing the mRNA and protein expression of VDR and its downstream molecules [34], thus alleviating the lung injury caused by sepsis [32]. In addition, emodin can bind to c-Jun N-terminal kinase (JNK2), inhibit the activation of NF-κB signaling [35], induce a protective effect against sepsis-associated intestinal mucosal barrier injury, increase the expression of tyrosine kinase receptor B (TrkB) and brain-derived neurotrophic factor (BDNF), and significantly inhibit the inflammatory response in CLP mice, thereby improving cognitive impairment and reducing pathological damage [36].

Aloin (CAS: 1415-73-2) is an anthraquinone compound with antitumor, anti-inflammatory, antiosteoporosis, antiviral, antibacterial, and other pharmacological properties [37]. As shown in Figure 5, aloin treatment can alleviate LPS-induced inflammation by inhibiting NF-κB signaling; blocking the phosphorylation, acetylation, and nuclear transport of p65 subunits; and downregulating stress-related genes [38,39]. At the same time, aloin significantly inhibits the activation of NLRP3 inflammatory bodies to improve LPS-induced acute lung injury and increase the expression of SIRT1 [40]. In LPS cell models and CLP-induced sepsis mouse models, deacetylation of HMGB1 achieved by activating SIRT1 reduces the release of HMGB1 and sepsis-related mortality [41]. Treatment with aloin has been shown to significantly reduce the levels of harmful renal functional substances, such as urea, creatinine, and urinary protein, and protect mice from sepsis-induced acute kidney injury [42]. Therefore, aloin, as a small-molecule drug, is effective in alleviating sepsis-induced acute kidney and lung injury.

## 4. Glucoside Compounds

Glycosides connect the end-group carbon atoms of sugars or sugar derivatives with non-sugar substances. Currently, the main glycosides used for treating sepsis are salidroside and geniposide.

Salidroside (CAS: 10338-51-9) is a constituent of herbal *Rhodiola rosea* L., extensively employed in the adjunctive therapy of cardiovascular and cerebrovascular disorders and certain neoplasms [43]. The use of salidroside in sepsis treatment is still being investigated through in vivo and in vitro experiments, and the drug has not yet entered clinical research. As shown in Figure 6, salidroside inhibits the production of caspase-3/9 by upregulating Bcl-2 and downregulating Bax, thereby inhibiting the phosphorylation of Phosphoinositide-3 kinase (PI3K) and AKT and reducing the levels of pro-inflammatory cytokines and apoptosis [44]. Salidroside can significantly reduce the expression of p65 in kidney tissue, reduce the levels of pro-inflammatory factors in the plasma and kidney, and alleviate sepsis-induced acute kidney injury in CLP models. It has also been shown to significantly reduce the mortality of septic rats [45]. Salidroside can enhance the expression of PPP1R15A and downregulate endoplasmic reticulum stress-related proteins, thereby inhibiting endoplasmic reticulum stress and improving lung injury in septic mice [46]. Salidroside can also inhibit the phosphorylation of NF-κB and PI3K/AKT/mTOR, significantly reduce the expression levels of ROS, CAT, SOD, GSH-px, TNF-α, IL-6, and IL-1β in cells, and have obvious cardioprotective effects on LPS-treated rats [47]. Thus, further investigation of the clinical applications of salidroside will be beneficial.

As a new iridoid glycoside, geniposide (CAS: 24512-63-8) has many biological activities, such as anti-inflammation, anti-oxidation, and anti-apoptosis properties [48]. As shown in Figure 7, in a mouse model of septic myocardial dysfunction induced by LPS, geniposide can activate AMPKα and inhibit myocardial reactive oxygen species (ROS) production, block NLRP3-mediated cardiomyocyte apoptosis and pyrolysis, and improve septic-induced myocardial dysfunction [49]. Additionally, in LPS-induced cell and CLP-induced sepsis mouse models, geniposide significantly inhibits the inflammatory response, apoptosis, oxidative stress, and vascular permeability associated with sepsis-induced acute kidney injury by activating Peroxisome proliferator-activated receptor γ(PPARγ) [50]. There are few reports of the use of geniposide in sepsis treatment, and its specific mechanism in this context requires further study.

## 5. Sterol Compounds

Sterols, also known as steroids, are lipid compounds. Currently, ginsenosides are the main steroids used for sepsis treatment. They are used in LPS- and CLP-induced sepsis models.

Ginsenosides are also known as triterpenoid saponins and are divided into many categories. Among them are Rb1 (CAS: 41753-43-9) [51], Rb3 (CAS: 68406-26-8) [52,53], Rd (CAS: 52705-93-8) [54], Re (CAS: 52286-59-6) [55], Rg1 (CAS: 22427-390) [56,57], Rg5 (CAS: 186763-78-0) [58], Rg6 (CAS: 147419-93-0) [59], and Rh1 (CAS: 63223-86-9) [60] reduce the expression of pro-inflammatory cytokines through the TLR4/NF-κB/MAPK signaling pathway to reduce inflammation and organ damage and improve the survival rate.

Furthermore, as shown in Figure 8, Rh1 can potentially attenuate the activation of TNF-α and IL-6 mediated by HMGB6 [61] and minimize tissue damage. Additionally, Rh1 and Rg2 can inhibit the production of mitochondrial reactive oxygen species (mtROS) [62], and Rg3 can activate the AMPK signaling pathway to promote mitochondrial autophagy, maintain mitochondrial homeostasis, and alleviate the subsequent inflammatory response, thereby reducing the cell and organ damage caused by sepsis and increasing the survival rate of septic rats [63]. Rg4 can activate PI3K/p-AKT signal transduction, inhibit the septic kidney inflammation induced by CLP, and improve survival [64]. In clinical treatment, the combination of total ginsenosides and ulinastatin has been shown to be effective against septic acute lung injury (ALI) and acute respiratory distress syndrome (ARDS) [65]. Thus, ginsenosides have great potential for the prevention and treatment of sepsis.

## 6. Flavonoid Compounds

Flavonoids are compounds resulting from the linkage of two benzene rings with three carbon atoms. The main flavonoids used in sepsis treatment are breviscapine, baicalein, and diosmtin.

As shown in Figure 9, breviscapine (BS, CAS: 116122-36-2) inhibits the overactivation of the TLR4/NF-κB, caspase-3/PARP, and MAPK signaling pathways, thereby inhibiting the expression of proinflammatory cytokines and chemokines [66]. Breviscapine can also regulate the PI3K/Akt/glycogen synthase kinase-3β (GSK-3β) pathway and inhibit myocardial inflammation and apoptosis of coronary microembolization (CME) to achieve cardiac protection [67].

Baicalein (CAS: 491-67-8) is an important component extracted from *Scutellaria baicalensis*. The use of baicalein in the treatment of sepsis has not yet been investigated clinically. As shown in Figure 10, baicalein can activate the AMPK pathway and inhibit downstream MAPK/NF-κB signal transduction and chemokines to inhibit the expression of ROS [68,69], thus inhibiting the activation of NLRP3 inflammatory bodies [70]. Baicalein inhibits the expression of dynamic protein-associated protein 1 (Drp1), reduces the levels of ROS, and reduces the production of TNF-α, MIP-1, and IL-6 to inhibit LPS-induced acute lung injury [71]. Moreover, baicalein can improve the sepsis-induced liver injury induced by LPS and CLP in septic mice by activating Nrf2 signaling in hepatocytes, which regulates antioxidation and pro-inflammatory signal transduction [72]. The above-mentioned findings suggest that baicalein may be a candidate drug for treating sepsis.

Diosmtin (Dio, CAS: 520-27-4) has anti-inflammation, anti-oxidation, and anti-apoptosis properties [73]. It is still being studied in the laboratory as a treatment for sepsis. As shown in Figure 11, in an LPS-induced cell model, Dio can alleviate sepsis-induced acute kidney injury by enhancing the activity of the Nrf2 pathway, increasing the expression of lncRNA-TUG1, and inhibiting the expression of caspase-3 [74]. In addition, vanilla lignin can activate Nrf2 signaling to clear ROS and inhibit the activation of NLRP3 to limit the development of inflammation, which can alleviate LPS-induced acute lung injury [75].

## 7. Biogenic Amines

Biogenic amines, which are organic compounds with low molecular weights, possess biological activity and have a high nitrogen content.

Agmatine (AGM, CAS: 306-60-5) is a naturally occurring polyamine synthesized by the enzyme L-arginine decarboxylase within the central nervous system. AGM is broadly distributed within the liver and central nervous system [76]. As shown in Figure 12, agmatine alleviates vascular dysfunction in LPS-treated rats by inhibiting the expression of inducible nitric oxide synthase (iNOS) and oxidative stress [77]. Agmatine can also inhibit the phosphorylation and degradation of IκB, thereby inhibiting the activation of NF-κB signal transduction and reducing systemic inflammation and organ failure in LPS mice [78].

## 8. Alkaloid Compounds

Alkaloids are a class of basic organic compounds containing nitrogen. Most alkaloids have a complex ring structure and a range of biological activities. The main alkaloids used for the prevention and treatment of sepsis are kukoamine B, matrine, anisodamine hydrobromide, berberine, and leonurine.

As shown in Figure 13, kukoamine B (CAS: 164991-67-7), as a novel dual inhibitor of LPS and CpG DNA, regulates the downstream signal pathway by directly binding and neutralizing LPS and CpG DNA, thus significantly inhibiting the inflammatory response in LPS-induced septic mice [79]. Wang et al. [80] used an exposure-response model to optimize dose selection in phase IIb clinical trials and recommended a 0.24 mg/kg regimen. A randomized, double-masked, placebo-controlled, multi-dose phase I study also demonstrated that single and multiple intravenous infusions of 0.06–0.24 mg/kg were safe and tolerable in healthy volunteers [81].

Matrine (CAS: 519-02-8) is the main alkaloid used in traditional Chinese herbal medicine and is extracted from *Sophora flavescens* (Leguminosae) [82]. Matrine has many biological activities, such as anti-tumor, anti-inflammation, analgesia, anti-fibrosis, anti-viral, and anti-arrhythmia properties, and can enhance immune function [83]. Matrine is still being studied in the laboratory as a sepsis treatment. As shown in Figure 14, matrine has been shown to restore the levels of miR-9 reduced by LPS by inhibiting the JNK and NF-κB pathways, thereby protecting cells from tissue damage induced by LPS [82]. Matrine can inhibit the TLR4/MyD88/NF-κB pathway, NLRP3 inflammatory body activation, and the secretion of proinflammatory cytokines [84] and can regulate the JNK signaling pathway to inhibit the activation of NLRP2 inflammatory bodies, thereby effectively alleviating the symptoms of CLP-induced sepsis in mice [85]. Matrine can also downregulate PTENP1 and upregulate miR-106b-5p to enhance the vitality of cardiac myoblasts and reduce the inflammatory response, thus alleviating the cardiac insufficiency caused by sepsis [86].

Anisodamine hydrobromide (CAS: 55449-49-5) is an alkaloid first extracted from *Scopolia tangutica* root in 1956 [87]. Tablets and injections made from anisodamine hydrobromide are often used as anticholinergic drugs in the clinic. As shown in Figure 15, anisodamine hydrobromide inhibits NF-κB signaling, thereby inhibiting LPS-induced apoptosis and inflammation [88], and alleviates LPS-induced acute kidney failure by inhibiting mitochondrial dysfunction and oxidative stress [87]. At the same time, anisodamine hydrobromide can inhibit cell death and apoptosis by inhibiting the gasdermin D (GSDMD) pathway, thereby reducing acute lung injury [89]. In addition, anisodamine hydrobromide can impede the degradation process of vascular endothelial cadherin, thereby preserving the integrity of the vascular endothelial barrier and enhancing microcirculation [90].

Berberine (CAS: 2086-83-1) is the main bioactive component extracted from the bark of *Phellodendron chinensis* and *Coptis chinensis* in traditional Chinese medicine. Clinical adverse events are rarely reported [91]. Berberine has a therapeutic effect against cardiac dysfunction, myocardial injury, and intestinal vascular barrier dysfunction caused by sepsis. As shown in Figure 16, berberine increases the activity of total nitric oxide synthase (NOS) in the heart, increases the protein expression of p-Akt and phosphorylated endothelial NOS, decreases the expression of inflammatory factors such as TNF-α and IL-1β by inhibiting the activation of the TLR4/NF-κB signaling pathway, and alleviates the cardiac dysfunction and myocardial injury caused by sepsis in LPS rat and mouse models [92,93]. In addition, berberine exhibits a protective effect against the intestinal vascular barrier dysfunction induced by sepsis in both LPS cell models and CLP rat models, which is related to berberine-induced downregulation of Wnt/β-catenin signaling [94].

Leonurine (CAS: 24697-74-3) is a natural alkaloid with anti-inflammatory and antioxidant properties [95]. As shown in Figure 17, leonurine can alleviate LPS-induced myocarditis by inhibiting the expression of p-IκBα and p-p65 [96]. Furthermore, leonurine can mitigate the LPS-induced acute lung injury in mice by inhibiting oxidative stress and inflammation, which are regulated by the Nrf2 signaling pathway [95]. However, the application of leonurine in the treatment of sepsis is a relatively new area of research and has mostly been limited to experimental animal research.

## 9. Amide Compounds

Glutamine (CAS: 56-85-9) is a type of l-α-amino acid containing five kinds of carbon and is considered the most abundant amino acid in the human body. As a single nutrient supplement, a reasonable dose of glutamine is considered safe [97]. As shown in Figure 18, glutamine supplementation in the abdominal cavity can reduce sepsis-induced damage to the intestinal mucosa, kidney, and liver tissues in CLP rat models [98]. Glutamine can inhibit the expression of SIRT5 to inhibit the desuccinylation of pyruvate dehydrogenase (PDH), which leads to an increase in oxidative phosphorylation, thereby promoting M2 polarization of macrophages to reduce burn sepsis in mice [99]. In clinical trials, intravenous administration and enteral combined administration of glutamine effectively improved transferrin, creatine/height index, and nitrogen balance in patients with dystrophic sepsis, with the best effects on days 7 and 15 [100].

## 10. Discussion and Prospects

This review provides a comprehensive overview of the functions and mechanisms of natural small-molecule compounds in the prevention and treatment of sepsis. These natural small-molecule compounds can modulate many signaling pathways, such as NF-κB, TLR4, MAPK, NLRP3, AMPK, and PI3KAKT, to prevent and treat sepsis. Most of these compounds target proteins to inhibit pathways that cause inflammation; this prevents more serious organ damage and failure and can alleviate sepsis-related damage to the heart, liver, kidney, intestinal tract, and lung. However, the application of most natural small-molecule compounds in the treatment of sepsis is still in the early stages of laboratory research, and there are few reports on dose, administration time, treatment cycle, and adverse reactions. Therefore, many experimental studies are needed to further explore the clinical application of these compounds for treating sepsis in patients. While some small-molecule compounds, such as the alkaloid kukoamine B, have entered the clinical treatment stage, none have been approved or are widely used. Table 1 summarizes the experimental models reported in this paper for evaluating the use of small-molecule compounds in sepsis treatment, the improvements and adverse reactions in the experimental models, and the research stage.

Studying therapeutic drugs for sepsis has always been a hotspot in medical research. Owing to the in-depth study of the pathogenesis of sepsis in recent years, various medicinal drugs have continued to emerge. However, there is still a long way to go to find effective and safe drugs for sepsis treatment. To better serve the clinic, we must explore the specific molecular mechanisms underlying the effects of these compounds in the context of sepsis and identify any adverse reactions. We have reason to believe that with further elucidation of the molecular mechanism of the occurrence and development of sepsis, the in-depth study of animal models, and the development of more clinical trials, more and more natural small-molecule compounds and other types of drugs can be developed as effective drugs for the prevention and treatment of sepsis.

## Figures and Tables

**Figure 1 ijms-24-12732-f001:**
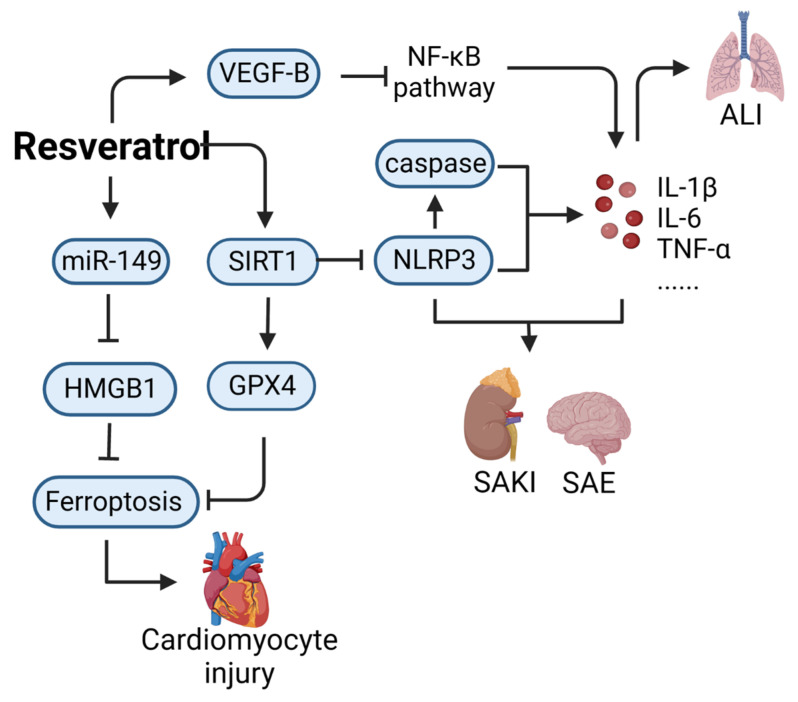
Action mechanism of resveratrol in the treatment of sepsis. ALI, acute lung injury; SAKI, sepsis-induced acute kidney injury; SAE, sepsis-associated encephalopathy; VEGF-B, vascular endothelial growth factor-B; SIRT1, Sirtuin1; NLRP3, NOD-like receptor protein 3; HMGB1, high mobility group protein B1; GPX4, glutathione Peroxidase 4.

**Figure 2 ijms-24-12732-f002:**
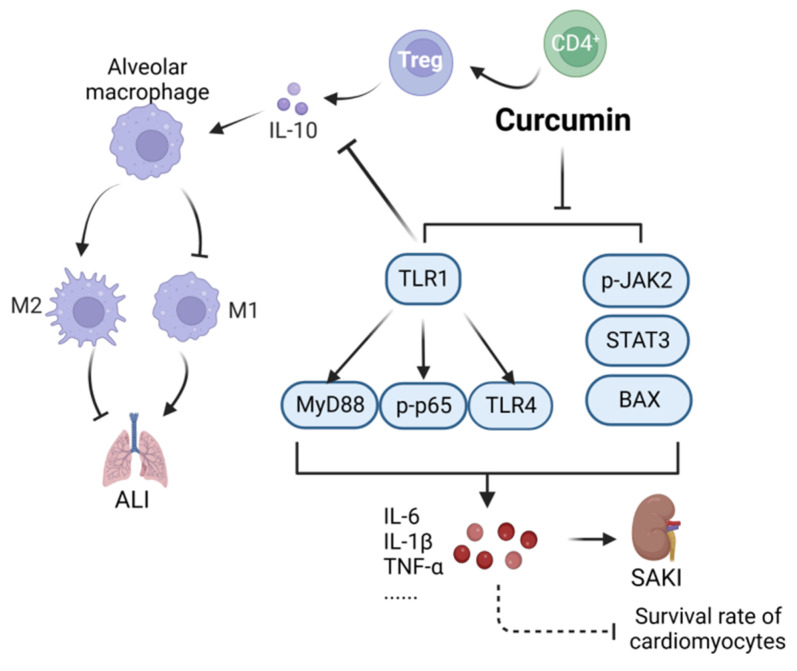
Action mechanism of curcumin in the treatment of sepsis. ALI, acute lung injury; SAKI, sepsis-induced acute kidney injury; MyD88, myeloid differentiation factor 88; BAX, BCL2-Associated X; STAT3, Signal Transducer and Activator of Transcription 3. Solid line: direct action; Dashed line: indirect action.

**Figure 3 ijms-24-12732-f003:**
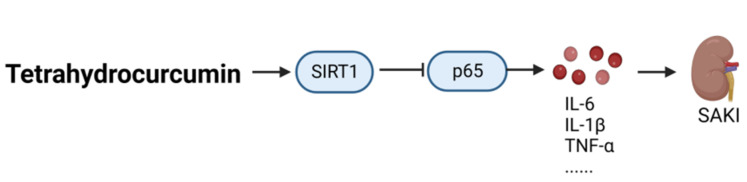
Therapeutic mechanism of tetrahydrocurcumin in the management of sepsis. SAKI, sepsis-induced acute kidney injury; SIRT1, Sirtuin1.

**Figure 4 ijms-24-12732-f004:**
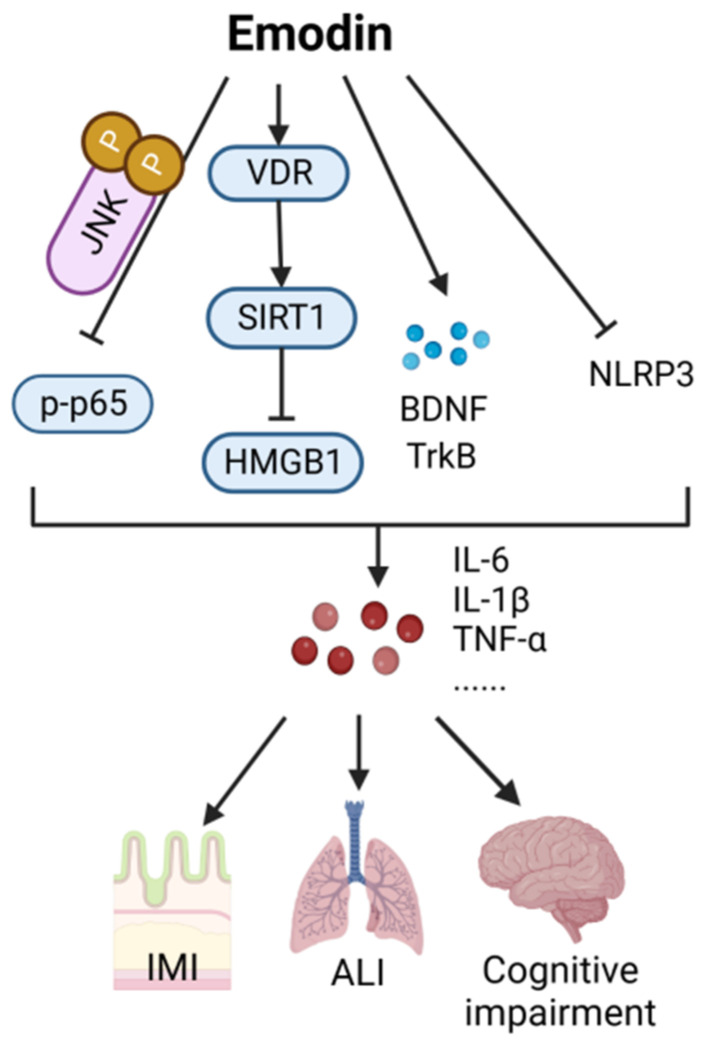
Action mechanism of emodin in the treatment of sepsis. IMI, intestinal mucosal barrier injury; ALI, acute lung injury; VDR, Vitamin D receptor; SIRT1, Sirtuin1; HMGB1, high mobility group protein B1; BDNF, brain-derived neurotrophic factor; TrkB, tyrosine kinase receptor B.

**Figure 5 ijms-24-12732-f005:**
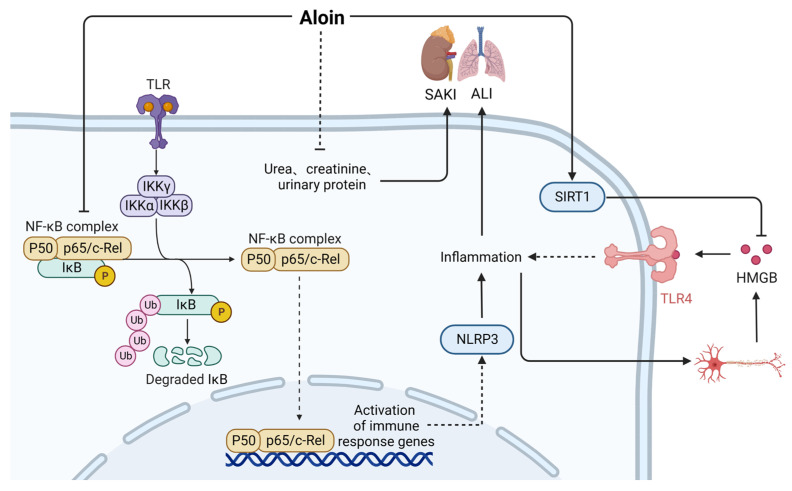
Action mechanism of aloin in the treatment of sepsis. SAKI, sepsis-induced acute kidney injury; ALI, acute lung injury; SIRT1, Sirtuin1; Solid line: direct action; Dashed line: indirect action.

**Figure 6 ijms-24-12732-f006:**
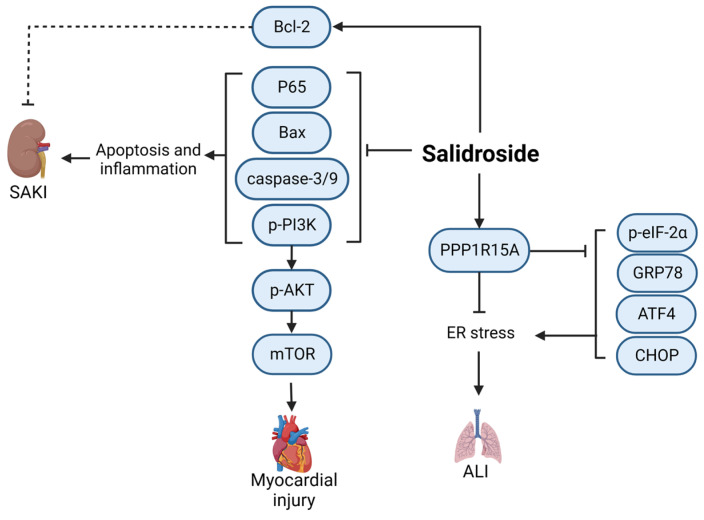
Mechanism of salidroside in the treatment of sepsis. ALI, acute lung injury; SAKI, sepsis-induced acute kidney injury; BAX, BCL2-Associated X; Bcl-2, B-cell lymphoma-2; mTOR, mammalian target of rapamycin; GRP78, glucose-regulated protein 78; ATF4, activating transcription factor 4; CHOP, C/EBP-homologous protein; Solid line: direct action; Dashed line: indirect action.

**Figure 7 ijms-24-12732-f007:**
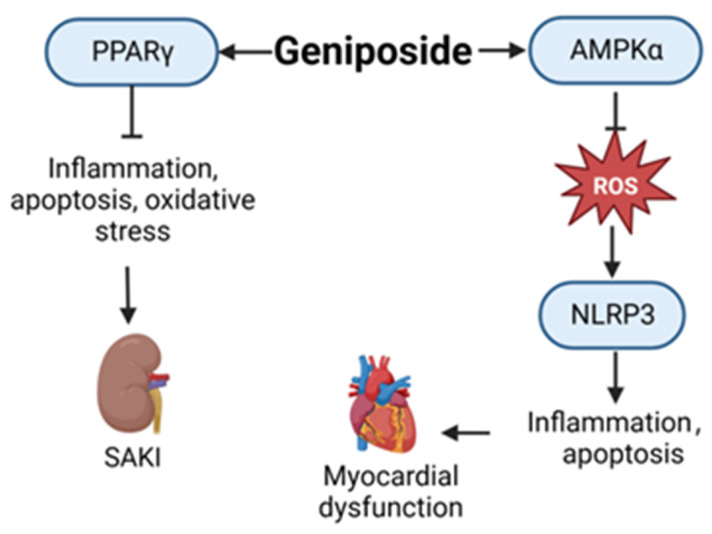
Action mechanism of geniposide in the treatment of sepsis. SAKI, sepsis-induced acute kidney injury; ROS, reactive oxygen species; PPARγ, Peroxisome proliferator-activated receptor γ.

**Figure 8 ijms-24-12732-f008:**
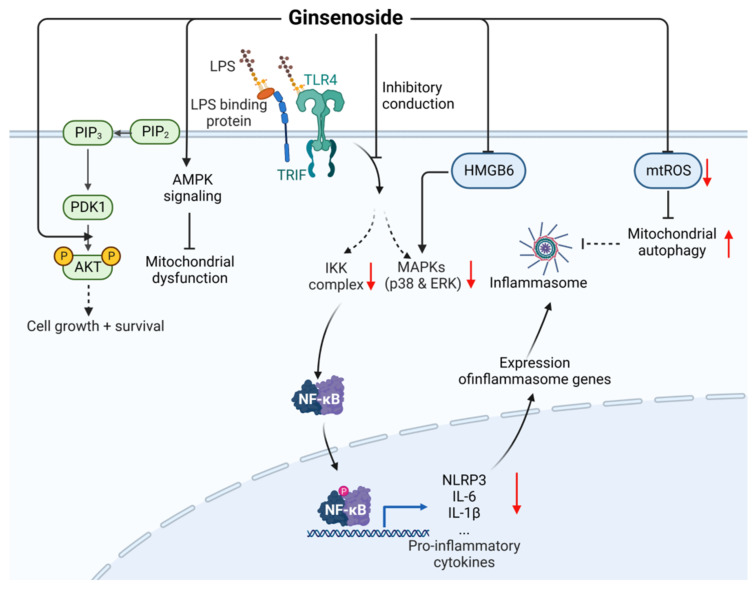
Mechanism underlying the therapeutic effects of ginsenoside against sepsis. LPS, lipopolysaccharides; HMGB6, high mobility group protein B6; mtROS, mitochondrial reactive oxygen species; Solid line: direct action; Dashed line: indirect action.

**Figure 9 ijms-24-12732-f009:**
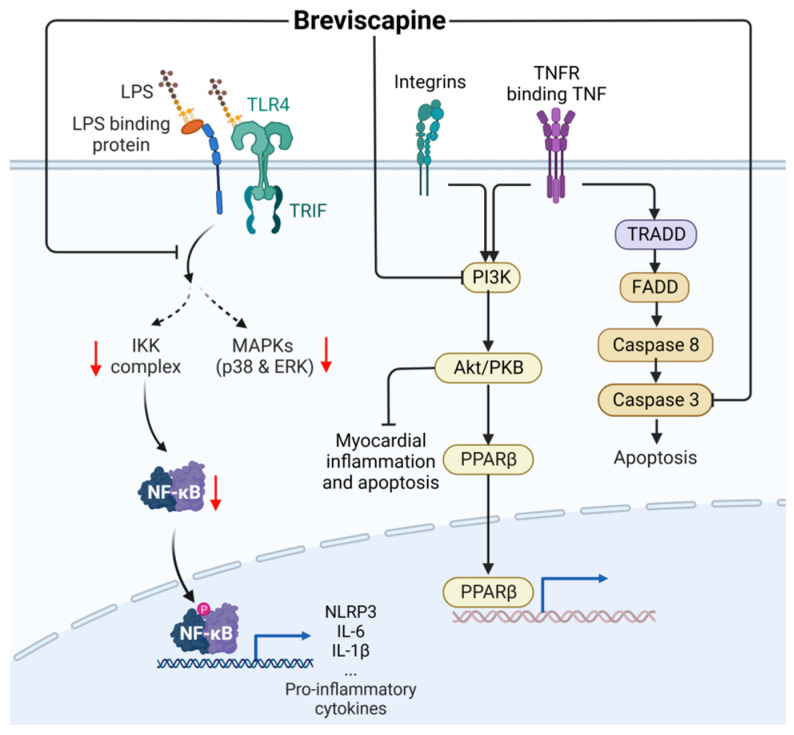
Action mechanism of breviscapine in the treatment of sepsis. LPS, lipopolysaccharides; TRADD, TNF receptor-associated death domain; FADD, fas-associated protein with death domain; PPARβ, Peroxisome proliferator-activated receptor β. Solid line: direct action; Dashed line: indirect action.

**Figure 10 ijms-24-12732-f010:**
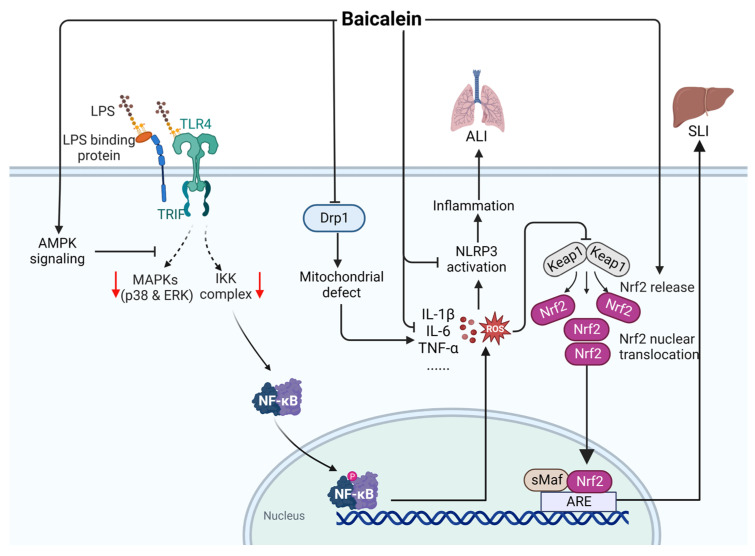
Action mechanism of baicalein in the treatment of sepsis. ALI, acute lung injury; SLI, sepsis-induced liver injury; LPS, lipopolysaccharides; Drp1, dynamic protein-associated protein 1; ROS, reactive oxygen species; Keap1, kelch like ECH associated protein 1; Nrf2, NF-E2-related factor 2; ARE, AU-rich element; sMaF, synthetic music mobile application format. Solid line: direct action; Dashed line: indirect action.

**Figure 11 ijms-24-12732-f011:**
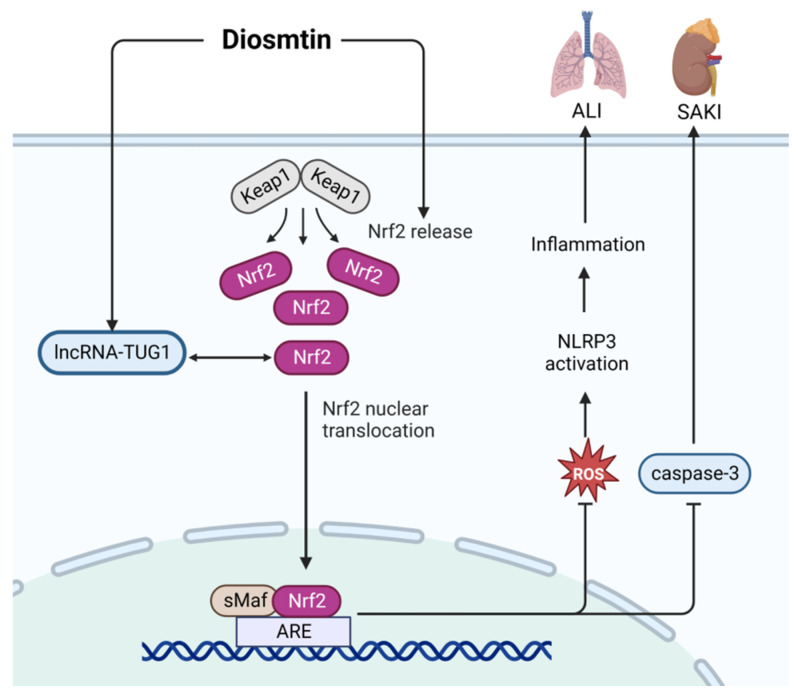
Action mechanism of diosmtin in the treatment of sepsis. ALI, acute lung injury; SAKI, sepsis-induced acute kidney injury; TUG1, Taurine Up-Regulated 1; Keap1, kelch like ECH associated protein 1; Nrf2, NF-E2-related factor 2; ARE, AU-rich element; sMaF, synthetic music mobile application format; ROS, reactive oxygen species. Solid line: direct action; Dashed line: indirect action.

**Figure 12 ijms-24-12732-f012:**
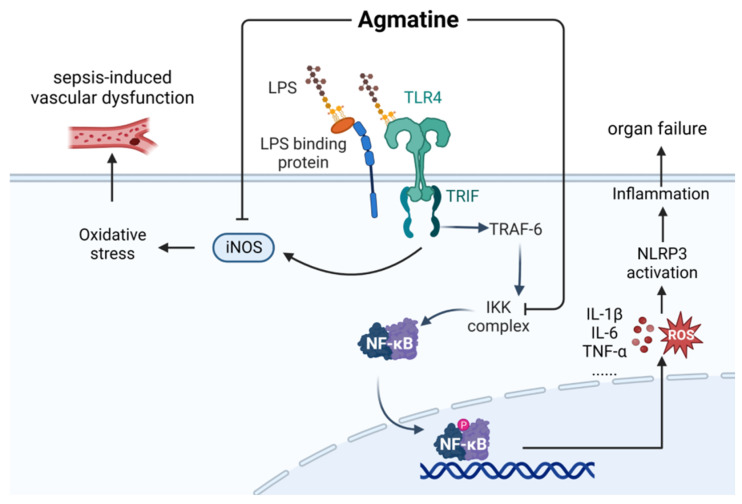
Mechanism underlying the therapeutic effects of agmatine in the management of sepsis. LPS, lipopolysaccharides; ROS, reactive oxygen species; iNOS, inducible nitric oxide synthase.

**Figure 13 ijms-24-12732-f013:**
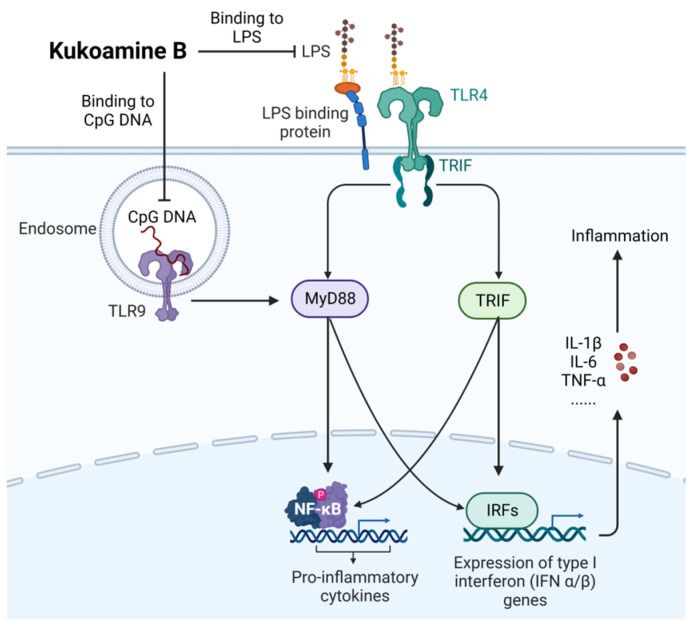
Action mechanism of kukoamine B in the treatment of sepsis. LPS, lipopolysaccharides; MyD88, myeloid differentiation factor 88.

**Figure 14 ijms-24-12732-f014:**
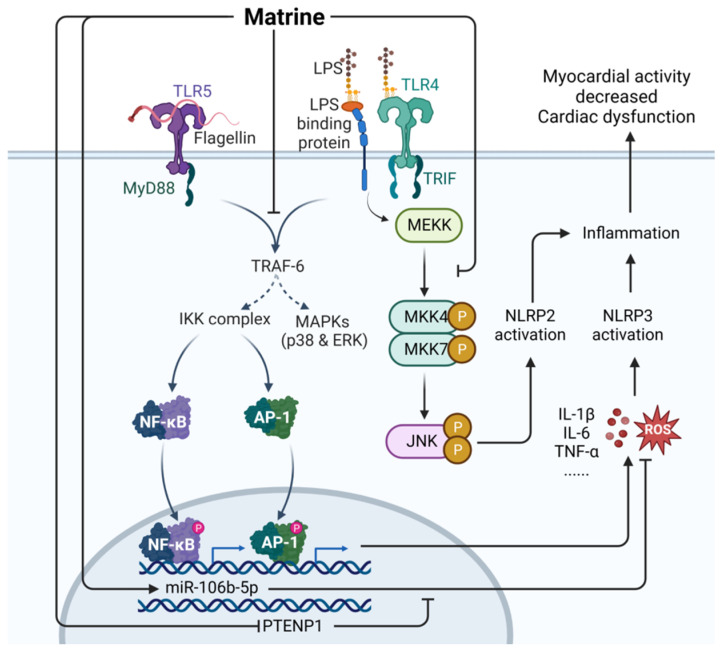
Mechanism of matrine in the treatment of sepsis. LPS, lipopolysaccharides; PTENP1, phosphatase and tensin homolog pseudogene 1; MyD88, myeloid differentiation factor 88; ROS, reactive oxygen species; Solid line: direct action; Dashed line: indirect action.

**Figure 15 ijms-24-12732-f015:**
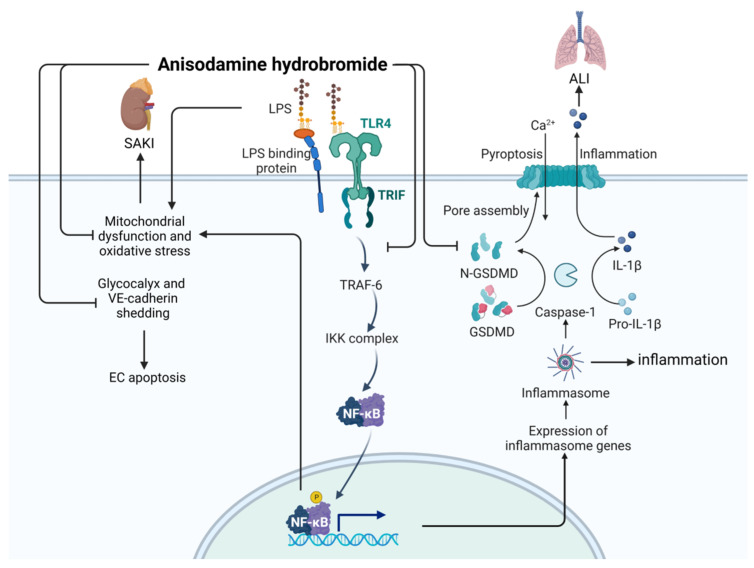
Action mechanism of anisodamine hydrobromide in the treatment of sepsis. ALI, acute lung injury; SAKI, sepsis-induced acute kidney injury; LPS, lipopolysaccharides; GSDMD, gasdermin D.

**Figure 16 ijms-24-12732-f016:**
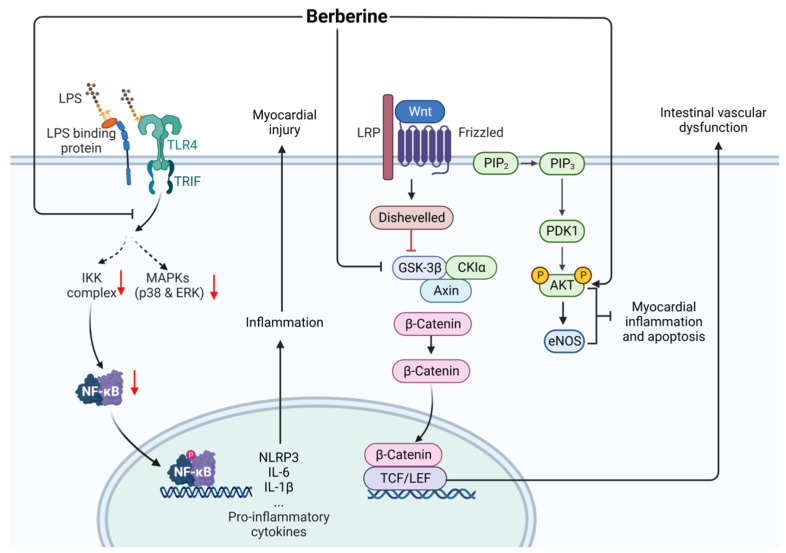
Action mechanism of berberine in the treatment of sepsis. LPS, lipopolysaccharides; LRP, lung resistance protein; eNOS, endothelial nitric oxide; Solid line: direct action; Dashed line: indirect action.

**Figure 17 ijms-24-12732-f017:**
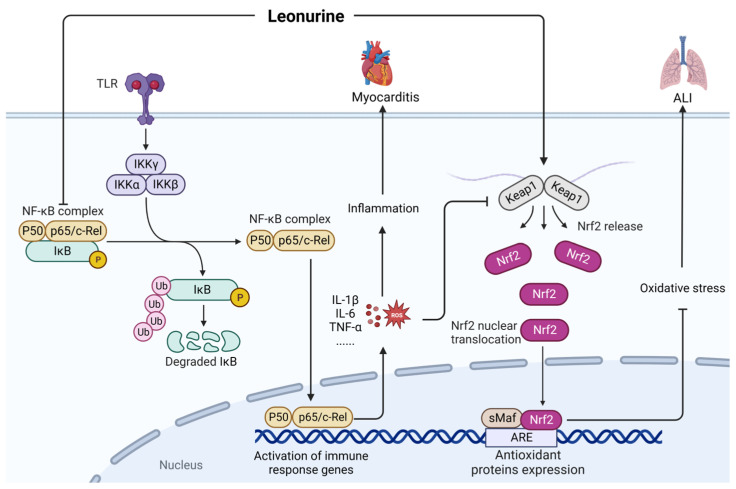
Mechanism of leonurine in the treatment of sepsis. ALI, acute lung injury; ROS, reactive oxygen species; Keap1, kelch like ECH associated protein 1; Nrf2, NF-E2-related factor 2; ARE, AU-rich element; sMaF, synthetic music mobile application format.

**Figure 18 ijms-24-12732-f018:**
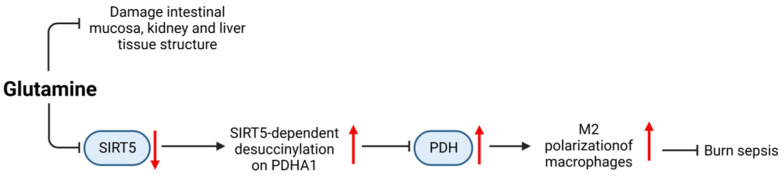
Mechanism of glutamine in the treatment of sepsis.

**Table 1 ijms-24-12732-t001:** Natural small-molecule drugs for sepsis treatment.

Compound	Model	Improvement	Adverse Reaction	Research Progress
Resveratrol	CLP/LPS	SAKI/SAE/ALI/cardiomyocyte injury	High doses can increase intracellular oxidation	Animal experiment
Curcumin	CLP/LPS	SAKI/ALI		Animal experiment
Tetrahydrocurcumin	CLP	SAKI		Animal experiment
Emodin	CLP/LPS	ALI/IMI/cognitive dysfunction		Animal experiment
Aloin	CLP/LPS	SAKI/ALI		Animal experiment
Salidroside	CLP/LPS	SAKI/ALI/myocardial injury		Animal experiment
Geniposide	CLP/LPS	SAKI/myocardial dysfunction		Animal experiment
Ginsenoside	CLP/LPS	Inflammatory response and organ damage		Animal experiment
Breviscapine	CME	Myocardium inflammation		Animal experiment
Baicalein	CLP/LPS	ALI/SLI		Animal experiment
Diosmtin	LPS	SAKI/ALI		Cell experiment
Agmatine	LPS	Vascular dysfunction/systemic inflammation and organ failure		Animal experiment
Kukoamine B	LPS	Inflammation		clinical trials
Matrine	CLP/LPS	Cardiac insufficiency		Animal experiment
Anisodamine hydrobromide	LPS	SAKI/ALI		Animal experiment
Berberine	CLP/LPS	Cardiac dysfunction, myocardial injury, and intestinal vascular barrier dysfunction		Animal experiment
Leonurine	LPS	ALI/myocarditis		Animal experiment
Glutamine	CLP	Damage to the intestinal mucosa, kidney, and liver tissues		Clinical trials

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
