# Peer review of "Research Progress on Natural Small-Molecule Compounds for the Prevention and Treatment of Sepsis"

_ijms, 2023, doi:10.3390/ijms241612732_

Round 1

Reviewer 1 Report

Su et al have discussed the mechanisms of natural small molecules compounds in preventing and treating sepsis.

Specific Comments:

1.)   Curcumin is extensively studied molecule of turmeric (Curcuma longa) but there are other molecules such as Turmerones and non-carbonyl fraction of Curcuma Oil have been shown to treat inflammation e.g. LPS-induced endotoxemia. Please also mention the mechanisms and the clinical trials on curcumin and turmerones.

2.)   Mechanisms of each compound has been shown but it is not clear which model of sepsis was used. Although, it is mentioned for some of the molecules but please mention which model (e.g. CLP, CASP, LPS or Pneumonia) was used for all the other molecules. As, each model has its own mechanism of action.

3.)   There is only one molecule (kukoamine B) has been shown for clinical trial. Please also mention the human studies of other molecules.

Author Response

Responses to the comments of Reviewer #1

  1. Curcumin is extensively studied molecule of turmeric (Curcuma longa) but there are other molecules such as Turmerones and non-carbonyl fraction of Curcuma Oil have been shown to treat inflammation e.g. LPS-induced endotoxemia. Please also mention the mechanisms and the clinical trials on curcumin and turmerones.

Response:

We wish to extend our sincere gratitude to the reviewer for their invaluable suggestions.

In accordance with the recommendations of the reviewer, the text has been enhanced with supplementary information regarding turmerones. For further elucidation on the modification, please refer to the details provided herein (Line 97-102):

Curcuma longa extract is rich in turmeric. The main compound present in turmeric is β-turmerone (CAS:19693-54-0). In a previous study, treatment with Curcuma longa extract had an anti-inflammatory effect and reduced the production of NO in an inflammation model induced by LPS [27]. In another study, targeted inhibition of TLR4 mediated the downstream information transmission, thereby effectively preventing the brain injury caused by neuroinflammation in LPS model mice [28].

  1. Mechanisms of each compound has been shown but it is not clear which model of sepsis was used. Although, it is mentioned for some of the molecules but please mention which model (e.g. CLP, CASP, LPS or Pneumonia) was used for all the other molecules. As, each model has its own mechanism of action.

Response:

We would like to extend our heartfelt appreciation for the invaluable suggestion put forth by the reviewer.

The utilization of in vitro and in vivo approaches in sepsis research has involved the use of LPS and CLP models. Relevant contents of sepsis model have been added in the paper, as follows:

  • Line 94-97

Curcumin has also been reported to inhibit NF-κB and JAK2/STAT3 signaling and the expression of p-JAK2/STAT3, pp65, and BAX in mice with acute kidney injury to alleviate septic acute kidney injury effectively in CLP mouse models [26].

  • Line 110-112

As shown in Figure 3, tetrahydrocurcumin significantly increased the expression of SIRT1 and inhibited inflammation and oxidative stress, thereby preventing sepsis-induced acute kidney injury in a CLP mouse model [29].

  • Line 147-149

In LPS cell models and CLP-induced sepsis mouse models, deacetylation of HMGB1 achieved by activating SIRT1 reduces the release of HMGB1 and sepsis-related mortality [41]

  • Line 168-171

Salidroside can significantly reduce the expression of p65 in kidney tissue, reduce the lev-els of pro-inflammatory factors in the plasma and kidney, and alleviate sepsis-induced acute kidney injury in CLP models.

  • Line 189-192

Additionally, in LPS-induced cell and CLP-induced sepsis mouse models, geniposide significantly inhibits the inflammatory response, apoptosis, oxidative stress, and vascular permeability associated with sepsis-induced acute kidney injury by activating PPARγ [50]

  • Line 200-202

Currently, ginsenosides are the main steroids used for sepsis treatment. They are used in LPS and CLP-induced sepsis models.

  • Line 227-232

Breviscapine can also regulate the PI3K/Akt/glycogen synthase kinase-3 β (GSK-3β) pathway and inhibit myocardial inflammation and apoptosis of coronary microemboli-zation (CME) to achieve cardiac protection [67].

  • Line 244-246

Moreover, baicalein can improve the sepsis-induced liver injury induced by LPS and CLP in septic mice by activating Nrf2 signaling in hepatocytes, which regulates antioxidation and pro-inflammatory signal transduction [72].

  • Line 254-257

As shown in Figure 11, in an LPS-induced cell model, Dio can alleviate sepsis-induced acute kidney injury by enhancing the activity of the Nrf2 pathway, increasing the expres-sion of lncRNA-TUG1, and inhibiting the expression of caspase-3 [74].

  • Line 273-276

Agmatine can also inhibit the phosphorylation and degradation of IκB, thereby inhibiting the activation of NF-κB signal transduction and reducing systemic inflammation and organ failure in LPS mice [78].

  • Line 332-339

As shown in Figure 16, berberine increases the activity of total nitric oxide synthase (NOS) in the heart, increases the protein expression of p-Akt and phosphorylated endothelial NOS, decreases the expression of inflammatory factors such as TNF- α and IL-1 β by inhibiting the activation of the TLR4/NF-κB signaling pathway, and alleviates the cardiac dysfunction and myocardial injury caused by sepsis in LPS rat and mouse models [92,93]. In addition, berberine exhibits a protective effect against the intestinal vascular barrier dysfunction induced by sepsis in both LPS cell models and CLP rat models, which is related to berberine-induced downregulation of Wnt/β-catenin signaling [94].

  • Line 345-347

Furthermore, leonurine can mitigate the LPS-induced acute lung injury in mice by inhib-iting oxidative stress and inflammation, which are regulated by the Nrf2 signaling path-way [95].

  • Line 355-357

As shown in Figure 18, glutamine supplementation in the abdominal cavity can reduce sepsis-induced damage to the intestinal mucosa, kidney, and liver tissues in CLP rat models [98].

3.There is only one molecule (kukoamine B) has been shown for clinical trial. Please also mention the human studies of other molecules.

Response:

We express our sincere gratitude for the reviewer's invaluable suggestion. The reports spanning from 2012 to 2023 were thoroughly examined, revealing that a significant proportion of the natural small molecule drugs discussed in the article have not been subjected to clinical trials for sepsis treatment as of yet. Kukoamine B and Ginsenosides have been reported in relevant clinical studies. In accordance with the viewpoint of the reviewer, the revised manuscript has incorporated the new sections as delineated below:

(1) Line 66-67

this compound has not yet entered the stage of clinical research regarding sepsis treat-ment.

(2) Line 94-97

Curcumin has also been reported to inhibit NF-κB and JAK2/STAT3 signaling and the expression of p-JAK2/STAT3, pp65, and BAX in mice with acute kidney injury to alleviate septic acute kidney injury effectively in CLP mouse models [26].

(3) Line 110-112

As shown in Figure 3, tetrahydrocurcumin significantly increased the expression of SIRT1 and inhibited inflammation and oxidative stress, thereby preventing sepsis-induced acute kidney injury in a CLP mouse model [29].

(4) Line 147-149

In LPS cell models and CLP-induced sepsis mouse models, deacetylation of HMGB1 achieved by activating SIRT1 reduces the release of HMGB1 and sepsis-related mortality [41]

(5) Line 164-165

The use of salidroside in sepsis treatment is still being investigated through in vivo and in vitro experiments, and the drug has not yet entered clinical research.

(6) Line 168-171

Salidroside can significantly reduce the expression of p65 in kidney tissue, reduce the lev-els of pro-inflammatory factors in the plasma and kidney, and alleviate sepsis-induced acute kidney injury in CLP models.

(7) Line 189-192

Additionally, in LPS-induced cell and CLP-induced sepsis mouse models, geniposide significantly inhibits the inflammatory response, apoptosis, oxidative stress, and vascular permeability associated with sepsis-induced acute kidney injury by activating PPARγ [50]

Line 200-202

Currently, ginsenosides are the main steroids used for sepsis treatment. They are used in LPS and CLP-induced sepsis models.

(8) Line 216-218

In clinical treatment, the combination of total ginsenosides and ulinastatin has been shown to be effective against septic acute lung injury (ALI) and acute respiratory distress syndrome (ARDS) [65].

(9) Line 227-232

Breviscapine can also regulate the PI3K/Akt/glycogen synthase kinase-3 β (GSK-3β) pathway and inhibit myocardial inflammation and apoptosis of coronary microemboli-zation (CME) to achieve cardiac protection [67].

(10) Line 238-239

The use of baicalein in the treatment of sepsis has not yet been investigated clinically.

(11) Line 244-246

Moreover, baicalein can improve the sepsis-induced liver injury induced by LPS and CLP in septic mice by activating Nrf2 signaling in hepatocytes, which regulates antioxidation and pro-inflammatory signal transduction [72].

(12) Line 254

It is still being studied in the laboratory as a treatment for sepsis.

(13) Line 254-257

As shown in Figure 11, in an LPS-induced cell model, Dio can alleviate sepsis-induced acute kidney injury by enhancing the activity of the Nrf2 pathway, increasing the expres-sion of lncRNA-TUG1, and inhibiting the expression of caspase-3 [74].

(14) Line 273-276

Agmatine can also inhibit the phosphorylation and degradation of IκB, thereby inhibiting the activation of NF-κB signal transduction and reducing systemic inflammation and organ failure in LPS mice [78].

(15) Line 299-300

Matrine is still being studied in the laboratory as a sepsis treatment.

(16) Line 332-339

As shown in Figure 16, berberine increases the activity of total nitric oxide synthase (NOS) in the heart, increases the protein expression of p-Akt and phosphorylated endothelial NOS, decreases the expression of inflammatory factors such as TNF- α and IL-1 β by inhibiting the activation of the TLR4/NF-κB signaling pathway, and alleviates the cardiac dysfunction and myocardial injury caused by sepsis in LPS rat and mouse models [92,93]. In addition, berberine exhibits a protective effect against the intestinal vascular barrier dysfunction induced by sepsis in both LPS cell models and CLP rat models, which is related to berberine-induced downregulation of Wnt/β-catenin signaling [94].

(17) Line 345-347

Furthermore, leonurine can mitigate the LPS-induced acute lung injury in mice by inhib-iting oxidative stress and inflammation, which are regulated by the Nrf2 signaling path-way [95].

(18) Line 355-357

As shown in Figure 18, glutamine supplementation in the abdominal cavity can reduce sepsis-induced damage to the intestinal mucosa, kidney, and liver tissues in CLP rat models [98].

Reviewer 2 Report

The article by Su et al. is well-written and has merit for publication in this journal.

Minor issues:

The authors should mention `treating sepsis in experimental animals` instead of treating sepsis only as it gives an impression that these compounds are already in use in clinics. 

I would like to see a table mentioning the sepsis model employed for studying these small molecules, any adverse reactions noticed,  and improvements observed in animals. Further, mention in the table whether any of these compounds are currently in clinical trials. 

The English language is fine.

Author Response

Responses to the comments of Reviewer #2

  1. The authors should mention `treating sepsis in experimental animals` instead of treating sepsis only as it gives an impression that these compounds are already in use in clinics.

Response:

We express our sincere gratitude for the reviewer's invaluable suggestion. According to the reviewer's opinion, the elucidation of the therapeutic impact on experimental animals was further enhanced. We have made some modifications in the revised manuscript as outlined below:

(1) Line 66-67

this compound has not yet entered the stage of clinical research regarding sepsis treat-ment.

(2) Line 164-165

The use of salidroside in sepsis treatment is still being investigated through in vivo and in vitro experiments, and the drug has not yet entered clinical research.

(3) Line 238-239

The use of baicalein in the treatment of sepsis has not yet been investigated clinically.

(4) Line 254

It is still being studied in the laboratory as a treatment for sepsis.

(5) Line 299-300

Matrine is still being studied in the laboratory as a sepsis treatment.

  1. I would like to see a table mentioning the sepsis model employed for studying these small molecules, any adverse reactions noticed, and improvements observed in animals. Further, mention in the table whether any of these compounds are currently in clinical trials. 

Response:

We would like to extend our heartfelt appreciation for the invaluable suggestion offered by the reviewer. According to the reviewer's comments, we added a table in the revised manuscript (Line 385), as shown below:

Table 1. Natural small-molecule drugs for sepsis treatment

Serial number

Compound

Model

Improvement

Adverse reaction

Research progress

1            

Resveratrol

CLP/LPS

SAKI/SAE/ALI/cardiomyocyte injury

High doses can increase intracellular oxidation

Animal experiment

2            

Curcumin

CLP/LPS

SAKI/ALI

Animal experiment

3            

Tetrahydrocurcumin

CLP

SAKI

Animal experiment

4            

Emodin

CLP/LPS

ALI/IMI/cognitive

dysfunction

Animal experiment

5            

Aloin

CLP/LPS

SAKI/ALI

Animal experiment

6            

Salidroside

CLP/LPS

SAKI/ALI/myocardial injury

Animal experiment

7            

Geniposide

CLP/LPS

SAKI/myocardial dysfunction

Animal experiment

8            

Ginsenoside

CLP/LPS

Inflammatory response and organ damage

Animal experiment

9            

Breviscapine

CME

Myocardium inflammation

Animal experiment

10        

Baicalein

CLP/LPS

ALI/SLI

Animal experiment

11        

Diosmtin

LPS

SAKI/ALI

Cell experiment

12        

Agmatine

LPS

Vascular dysfunction/systemic inflammation and organ failure

Animal experiment

13        

Kukoamine B

LPS

Inflammation

clinical trials

14        

Matrine

CLP/LPS

Cardiac insufficiency

Animal experiment

15        

Anisodamine hydrobromide

LPS

SAKI/ALI

Animal experiment

16        

Berberine

CLP/LPS

Cardiac dysfunction, myocardial injury, and intestinal vascular barrier dysfunction

Animal experiment

17        

Leonurine

LPS

ALI/myocarditis

Animal experiment

18        

Glutamine

CLP

Damage to the intestinal mucosa, kidney, and liver tissues

Clinical trials

Round 2

Reviewer 2 Report

The authors have addressed all my comments.